# Levofloxacin prophylaxis and parenteral nutrition have a detrimental effect on intestinal microbial networks in pediatric patients undergoing HSCT

Marco Fabbrini [1,2 ✉], Federica D'Amico[1,2], Davide Leardini[3], Edoardo Muratore [3], Monica Barone[1,2], Tamara Belotti[3], Maria Luisa Forchielli[4,5], Daniele Zama[3,5], Andrea Pession[3,5], Arcangelo Prete[3], Patrizia Brigidi[1], Simone Rampelli [2], Marco Candela [2], Silvia Turroni [2 ✉] & Riccardo Masetti[3,5]

The gut microbiome (GM) has shown to influence hematopoietic stem cell transplantation (HSCT) outcome. Evidence on levofloxacin (LVX) prophylaxis usefulness before HSCT in pediatric patients is controversial and its impact on GM is poorly characterized. Post-HSCT parenteral nutrition (PN) is oftentimes the first-line nutritional support in the neutropenic phase, despite the emerging benefits of enteral nutrition (EN). In this exploratory work, we used a global-to-local networking approach to obtain a high-resolution longitudinal characterization of the GM in 30 pediatric HSCT patients receiving PN combined with LVX prophylaxis or PN alone or EN alone. By evaluating the network topology, we found that PN, especially preceded by LVX prophylaxis, resulted in a detrimental effect over the GM, with low modularity, poor cohesion, a shift in keystone species and the emergence of modules comprising several pathobionts, such as *Klebsiella* spp., *[Ruminococcus] gnavus*, *Flavonifractor plautii* and *Enterococcus faecium*. Our pilot findings on LVX prophylaxis and PN-related disruption of GM networks should be considered in patient management, to possibly facilitate prompt recovery/maintenance of a healthy and well-wired GM. However, the impact of LVX prophylaxis and nutritional support on short- to long-term post-HSCT clinical outcomes has yet to be elucidated.

[1] Microbiomics Unit, Department of Medical and Surgical Sciences, University of Bologna, 40138 Bologna, Italy. [2] Unit of Microbiome Science and Biotechnology, Department of Pharmacy and Biotechnology, University of Bologna, 40126 Bologna, Italy. [3] Pediatric Oncology and Hematology Unit "Lalla Seràgnoli", IRCCS Azienda Ospedaliero-Universitaria di Bologna, 40138 Bologna, Italy. [4] Health Science and Technologies Interdepartmental Center for Industrial Research (CIRI-SDV), University of Bologna, 40100 Bologna, Italy. [5] Department of Medical and Surgical Sciences (DIMEC), University of Bologna, 40138 Bologna, Italy. ✉email: m.fabbrini@unibo.it; silvia.turroni@unibo.it

Hematopoietic stem cell transplantation (HSCT) is a potentially curative treatment for pediatric patients affected by a variety of hematological, oncologic, and immunologic diseases[1–4]. While often the only life-saving treatment, it is burdened by several life-threatening complications, including graft-versus-host disease and infections, which severely limit its broader applicability[5]. However, great strides have been made in the last decades, improving HSCT survival outcomes[6].

The gut microbiome (GM), i.e., the 10-trillion microbial community that inhabits our gut, is recognized as a key modifier of human health[7] and has already been extensively proven to impact HSCT complications and outcomes[8,9]. In particular, the available studies are consistent in highlighting a disruption of the GM structure and mutualistic framework after HSCT, which gradually recovers over time[10–12]. Alterations in reconstitution processes as well as distinct dysbiotic features during HSCT have been associated with various clinical outcomes, such as overall survival[13,14], relapse[15], immune reconstitution[16], graft-versus-host disease[17–19], infections[20] and other complications[21,22]. However, to date, the vast majority of works have focused on the mere compositional description of GM, in terms of diversity and variation of individual taxa, only sometimes coupled with functional data, i.e., the quantification of fecal short-chain fatty acids (SCFAs)[13,17,18,23–25]. Furthermore, there is a general lack of information on the combined effect on GM of some HSCT-related procedures, namely antibacterial prophylaxis and nutritional support. As for the former, this can be administered to reduce the incidence of infections, especially with the use of fluoroquinolones as levofloxacin (LVX)[26]. Evidence regarding the use of fluoroquinolone prophylaxis in adult neutropenic patients is controversial and the European Conference on Infections in Leukemia group (ECIL) recently suggested weighing the benefit of using fluoroquinolones in decreasing infection rates against the disadvantages of its toxicity and the dysbiotic changes induced in GM[27]. Also in the pediatric setting, recommendations are of low level of evidence, due to contradictory findings and the emergence of drug-related toxicity and resistant Gram-negative bacteria[28–33]. Regarding nutritional support, parenteral nutrition (PN) has historically been provided as first-line approach for nutritional support in the pediatric post-HSCT neutropenic phase but recent evidence has reported better outcomes of enteral nutrition (EN), in terms of lower risk of adverse effects, such as acute graft-versus-host disease, metabolic disorders and infectious complications[34–37], along with a faster recovery of a diverse GM profile, with greater SCFA production[25,38].

In this scenario, here we provide—for the first time to the best of our knowledge—a deep characterization of the ecology and functional structure of GM in pediatric HSCT patients in relation to antibiotic prophylaxis and nutritional support. Specifically, we performed whole-genome shotgun metagenomic sequencing of fecal samples collected from 30 patients receiving PN combined with LVX prophylaxis, or PN alone or EN alone, before and at 2 time points after transplantation, up to nearly 50 days later. We then applied a co-occurrence networking approach to shed light on the GM architecture and its dynamics. Unlike standard GM profiling methods, the application of network models based on shotgun metagenomics data allowed us to investigate species-level interactions and spot the most architecturally important members through the use of network metrics as hub scores and centrality measures. By taking into account the inherent connectivity structure of the ecosystem on a global and local scale, and by coupling such information with glimpses of functional emergent properties with respect to host interaction, we provided essential clues to disentangle the complex mechanisms underlying phenotypic manifestations, along with key accesses for potential interventions.

## Results

**Study cohort description.** Thirty pediatric patients undergoing HSCT from May 2015 to March 2019 at the Pediatric Transplant Unit of the IRCCS Azienda Ospedaliero-Universitaria di Bologna were enrolled as specified in the Methods. Patients were stratified according to prophylaxis administration and HSCT nutritional support as follows: (i) PN LVX (+), receiving PN combined with LVX prophylaxis ($n = 10$); (ii) PN LVX (−), receiving PN and not receiving LVX ($n = 10$); and (iii) EN, receiving EN and not receiving LVX ($n = 10$) (Supplementary Fig. 1). Clinical and transplant characteristics are reported in Supplementary Table 1. Detailed information regarding bloodstream infections and interim antibiotic administration is reported in Supplementary Table 2 and Supplementary Fig. 2, respectively. Fecal samples were collected at three timepoints: T0, before HSCT; T1, at engraftment; and T2, as the last follow-up before discharge.

**Global networks and biodiversity analysis elucidate distinct microbial profiles.** To unravel the dynamics of relationships among GM members, we used a co-occurrence networking approach based on Spearman's correlation matrices computed at species level[39]. First, we constructed an overall metapopulation network including all patient groups and timepoints, in order to fully represent the diversity of GM communities in our dataset (Fig. 1). On such a network, we identified three different community modules using statistical mechanics spin-glass model and simulated annealing[40].

One of the most striking findings is that *Klebsiella pneumoniae*, *Klebsiella quasipneumoniae* and *Klebsiella variicola* strongly and positively correlated with each other, forming a triad in module with other potentially pathobiont species, such as *[Ruminococcus] gnavus*, *Flavonifractor plautii*, *Escherichia coli*, *Enterococcus faecalis* and *Enterococcus faecium*, but firmly split off from the rest of the network, given the plethora of negative correlations against the other nodes. Such module was poorly populated in EN patients compared to both PN LVX (−) and PN LVX (+) patients, who showed higher relative abundances of the *Klebsiella* triad and members of its module.

The other two modules strongly and positively correlated with one another and included distinctive species of pediatric GM, mainly from the genus *Bifidobacterium*. Members of these modules were markedly underrepresented in the PN LVX (−) and especially PN LVX (+) group, with the latter showing a sharper reduction in the relative abundance of *Bifidobacterium* spp., *Ruthenibacterium lactatiformans*, *Fusicatenibacter saccharivorans* and *Faecalibacterium prausnitzii* compared with EN.

When evaluating alpha diversity with the Shannon index, we found a significant reduction related to LVX administration at all timepoints (Wilcoxon rank sum test with FDR correction, $p < 0.05$) (Fig. 2). In addition, at T1 the PN LVX (−) group was less diverse than the EN LVX (−) one ($p < 0.01$), suggesting a tangible impact of the parenteral feeding route as well. In order to assess the impact of interim antibiotics—aside from LVX prophylaxis—we investigated alpha diversity with respect to the number of different antibiotic molecules administered between the transplant date and each timepoint (Supplementary Fig. 3). No significant differences were observed (Kruskal–Wallis test, $p > 0.05$), suggesting that nutritional feeding and LVX prophylaxis might exert a greater effect over the GM than other interim antibiotics administered based on an empiric and pathogen-directed approach.

**Functional insights into microbial networks.** As shown in the heatmap of Fig. 3, the *Klebsiella* spp. triad showed a very strong positive correlation (Kendall $\tau > 0.9$) with pathways involved in

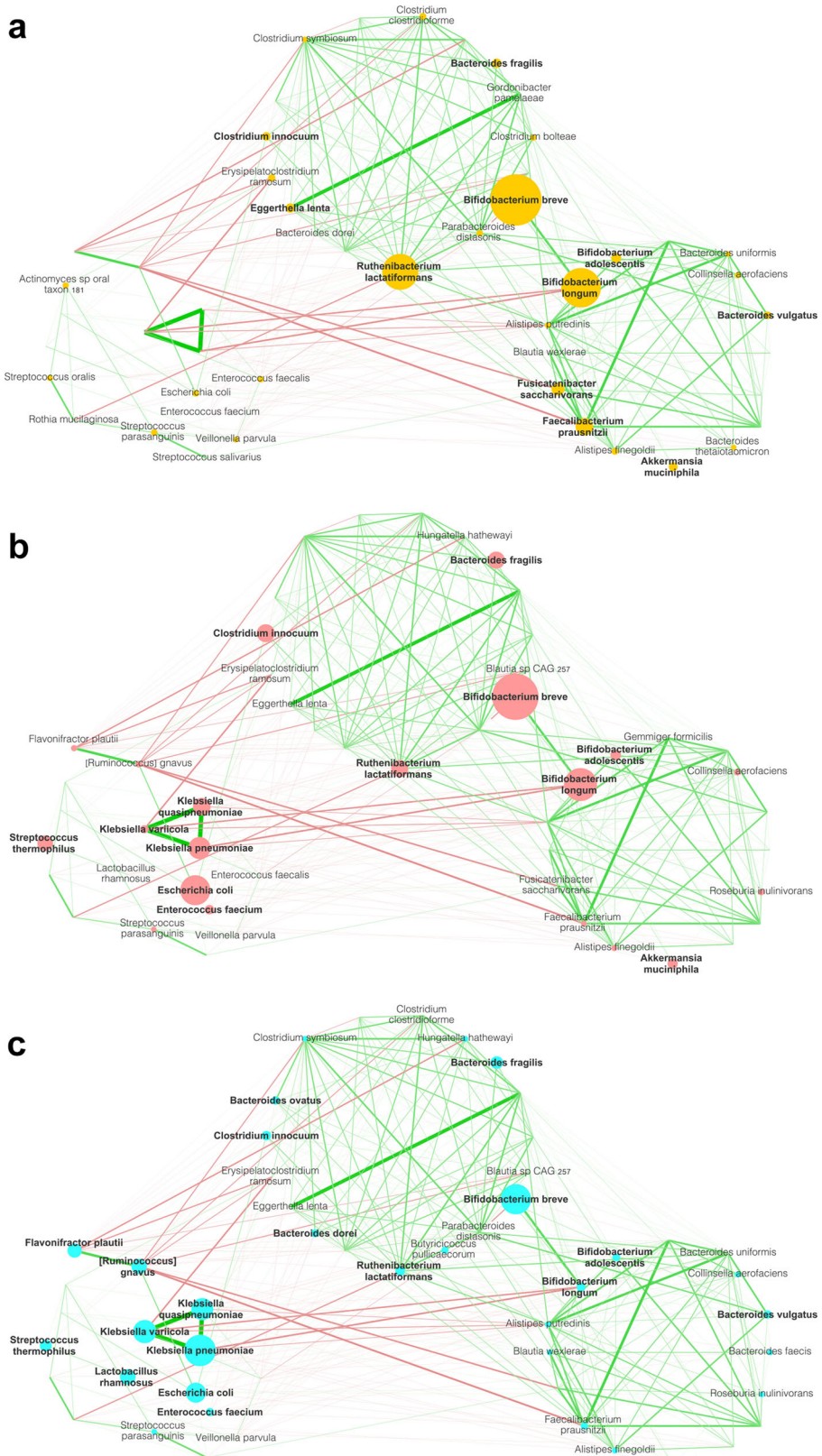

**Fig. 1 A different gut microbiome network structure is found in pediatric HSCT patients in association with EN, PN, and LVX prophylaxis.** Gut microbiome networks are shown for each treatment group (EN in yellow, (**a**); PN LVX (–) in pink, (**b**); PN LVX (+) in cyan, (**c**); *n* = *30* for each group), including all timepoints (i.e., before (T0) and after HSCT (T1 and T2)). Node sizes are proportional to the mean relative abundance of the species, and the network layout, derived from spin-glass clustering into modules, is maintained over the three plots. Green edges represent positive correlations, while red edges are indicative of negative correlations. Only edges with Spearman's correlation FDR-corrected *p* < 0.05 and (rho) *ρ* < −0.3 or *ρ* > 0.3 are shown. Edge line width is proportional to Spearman's *ρ*. To simplify the reading of the networks, only the nodes with an abundance >0.2% were labeled; bold labels correspond to species with an abundance higher than 1%. To construct the graph, only species with relative abundance >0.3% in at least 5% of the samples were considered.

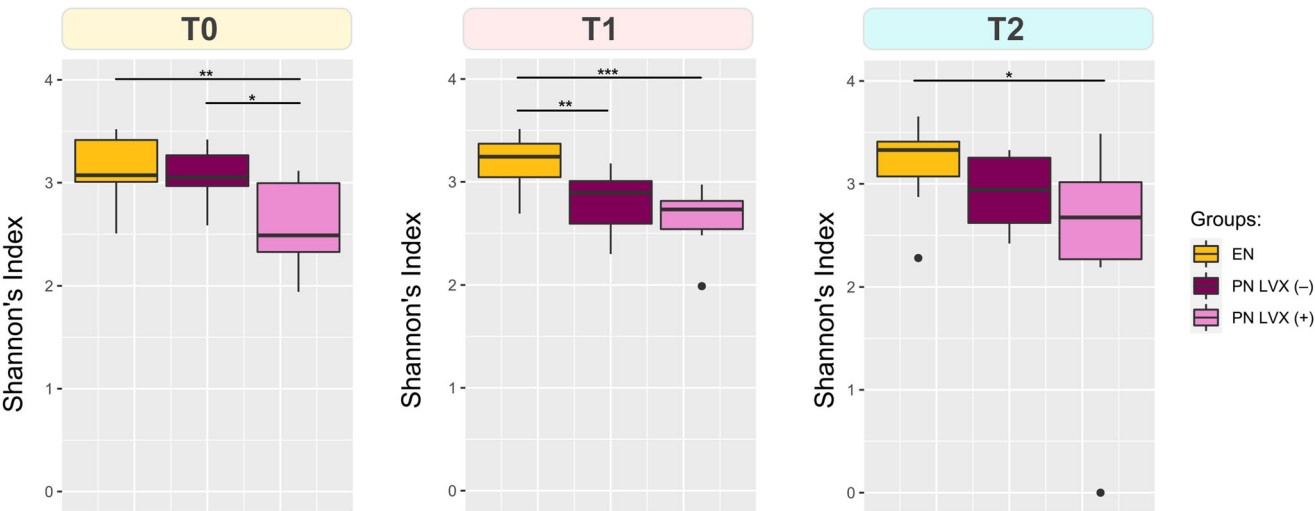

**Fig. 2 Levofloxacin prophylaxis results in a sustained reduction in alpha diversity.** Boxplots representing the distribution of alpha diversity estimated with the Shannon index for each patient group (EN, PN LVX (−), PN LVX (+)) before (T0) and at 2 time points after HSCT (T1 and T2). Significant differences between groups are shown (Wilcoxon rank sum test with FDR correction, *** $p < 0.001$, ** $p < 0.01$, * $p < 0.05$). $n = 10$ biologically independent samples for each timepoint. Whiskers represent range between the first quartile (Q1) and the third quartile (Q3). Data points outside the boundary of the whiskers are plotted as outliers.

the degradation of toluene and other aromatic compounds, i.e., 4-methyl-cathecol and protocatechuate. In addition, the triad was associated with L-ascorbate biosynthesis and, as might be expected, the enterobacterial common antigen biosynthesis pathway. On the other hand, species underrepresented in the PN-associated global networks as *Blautia wexlerae*, *F. prausnitzii*, *F. saccharivorans*, *Gemmiger formicilis* and *Ruminococcus bromii* were expectedly positively linked to pathways generally associated with the maintenance of microbiome-host mutualism, such as methanogenesis from acetate, starch degradation, biosynthesis of amino acids (arginine, lysine, branched-chain amino acids, and ornithine), and the methylerythritol phosphate (MEP) pathway involved in the biosynthesis of terpenes and terpenoids, molecules implicated in the reduction of intestinal inflammation and regulation of enterocyte permeability[41].

When focusing the analysis on the main metabolic classes, we found peculiar signatures of carbohydrate metabolism in the GM profile of PN LVX (−) patients, and xenobiotic metabolism in those associated with LVX prophylaxis (Fig. 4). In particular, a significantly higher xenobiotic degradation potential was found in the PN LVX (+) group at baseline compared to the EN and PN LVX (−) groups ($p < 0.01$), particularly for the toluene degradation pathway, which was ~10 times more represented in PN LVX (+) patients ($p < 0.001$). This is likely attributable to the fact that nearly all patients in the PN LVX (+) group had already been receiving LVX prophylaxis for at least 24 hours prior to T0 sampling (see Supplementary Fig. 1). In contrast, the chloroalkane and chloroalkene degradation pathway was significantly reduced before HSCT in patients undergoing prophylaxis ($p < 0.05$). Regarding carbohydrate metabolism, PN LVX (−) patients showed significantly higher metabolic capacity than EN patients at T0 and T2 (Wilcoxon rank sum test, $p < 0.05$), as well as compared to PN LVX (+) patients at T2 ($p < 0.01$).

**Hub microbes and keystone species**. To unravel the GM components driving the community structure in each network, we identified hub microbes as those taxa (i.e., those nodes) that were significantly more central in the network, based on three measures of centrality: degree (the number of direct connections to other nodes), betweenness centrality (quantifying the instances in

which a node is positioned on the shortest path between all the pairs of other nodes) and closeness centrality (an estimate of reciprocal node distances, which considers all the shortest paths between a given node and all the others) (see also Methods). To better elucidate network dynamics, we coupled this approach—purely based on network topology evaluation—with the identification of keystone species, i.e., nodes with a great quantity of overall interactions that depend on them. We decided to base the latter approach on the observation of the influence of each node on the total cohesion (TC), i.e., a metric of connectivity in terms of positive and negative interactions, specifically developed for microbial communities[42]. A keystone node was arbitrarily defined as a node whose absence caused a reduction greater than 50% in TC (i.e., more than half of the determining forces shaping the community state depended on its presence) (see also Methods). For a list of hub nodes and keystone species with the relative abundance and impact on TC values of each, respectively, see Table 1.

Interestingly, both hub microbes and keystone species were found to vary over time, but distinctly based on LVX prophylaxis and nutritional support. At baseline, we identified 5 keystone species in the GM network of EN patients, namely *Bifidobacterium breve*, *Bifidobacterium longum*, *R. lactatiformans*, *F. prausnitzii* and *Eggerthella lenta*, with the latter two also representing hub nodes. A general prevalence of hub species belonging to the genus *Bifidobacterium* (i.e., *B. longum* and *B. breve*) and phylum Bacteroidetes (i.e., *Bacteroides fragilis*, *Bacteroides uniformis* and *Parabacteroides distasonis*) was reported over time in the EN group. In particular, *B. longum* and *B. breve* were the most represented, with relative abundance values above 4%. Hub microbes *B. fragilis* and *B. longum* were also scored as keystones in EN-related networks at T1 and T2, respectively.

PN LVX (−) patients shared some hub nodes with EN but their baseline networks were mostly represented by *Akkermansia muciniphila*, followed by *Bifidobacterium adolescentis* (also scored as keystones). At T1, this group retained *E. lenta* among the hub nodes and exhibited *A. muciniphila* and *B. breve* as the only keystones. Then, at T2 their network was totally lacking in *Bifidobacterium* spp. as hubs but gained *Erysipelatoclostridium*

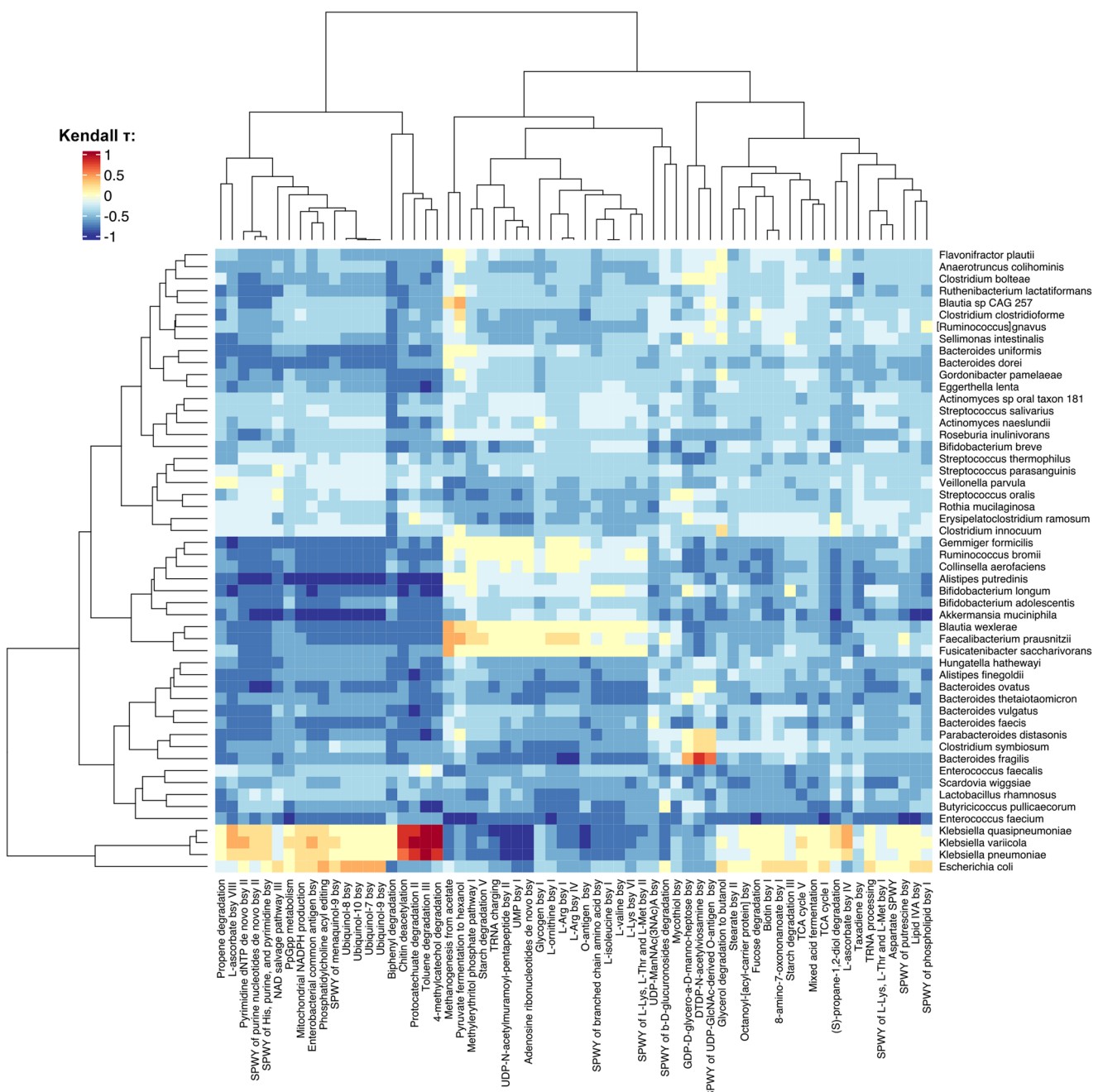

**Fig. 3 Functional potential of the components of the gut microbiome networks in pediatric HSCT patients.** Heatmap representing the Kendall correlation between species-level relative abundances and pathway CPMs (copies per million, see Methods) at all three timepoints. Species considered are the same used for network construction in Fig. 1. Only pathways that were differentially represented among groups ($p < 0.05$, Kruskal–Wallis test, grouping samples by feeding route, LVX administration, and timepoint) were considered. Pathway names (following the MetaCyc annotation) were shortened as follows: SPWY super pathway, bsy biosynthesis. $n = 90$ samples from 30 individuals.

*ramosum*. Other hubs at T2 included *B. uniformis*, also scored as a keystone together with *B. fragilis*.

Finally, PN LVX (+) patients showed the most distinct GM network structure: no EN-characteristic hubs were detected in this group, except for *C. innocuum* (but at T1 instead of T0) and *R. lactatiformans*, which was identified as a GM hub at T2 in all three patient groups, with almost comparable relative abundances (range, 3.35–3.55%). Considering hub nodes, at T0, the PN LVX (+) group featured *B. adolescentis* along with *F. plautii*, *F. saccharivorans* and *Roseburia inulivorans*, then it was discriminated by a number of potential pathobionts, such as *C. innocuum*, *K. quasipneumoniae* and *Clostridium symbiosum* at T1, and *R.*

*gnavus*, *F. plautii* and *E. faecium* at T2, and totally lacked *Bifidobacterium* spp. Notably, no keystone species was detected at any timepoint for PN LVX (+) patients based on the −50% impact threshold on TC. The highest value of TC reduction was found with the removal of *R. gnavus* (−13.87%) at T0, *A. muciniphila* (−25.5%) and the triad of *K. pneumoniae* (−14.99%), *K. quasipneumoniae* (−13.49%) and *K. variicola* (−13.85%) at T1, and again the *Klebsiella* triad (average −22.44%) at T2.

Compositional insights at family and species level showed concordant results with the networking approach and are reported in Supplementary Fig. 4. In particular, compared to LVX (−) patients (regardless of post-HSCT nutritional support),

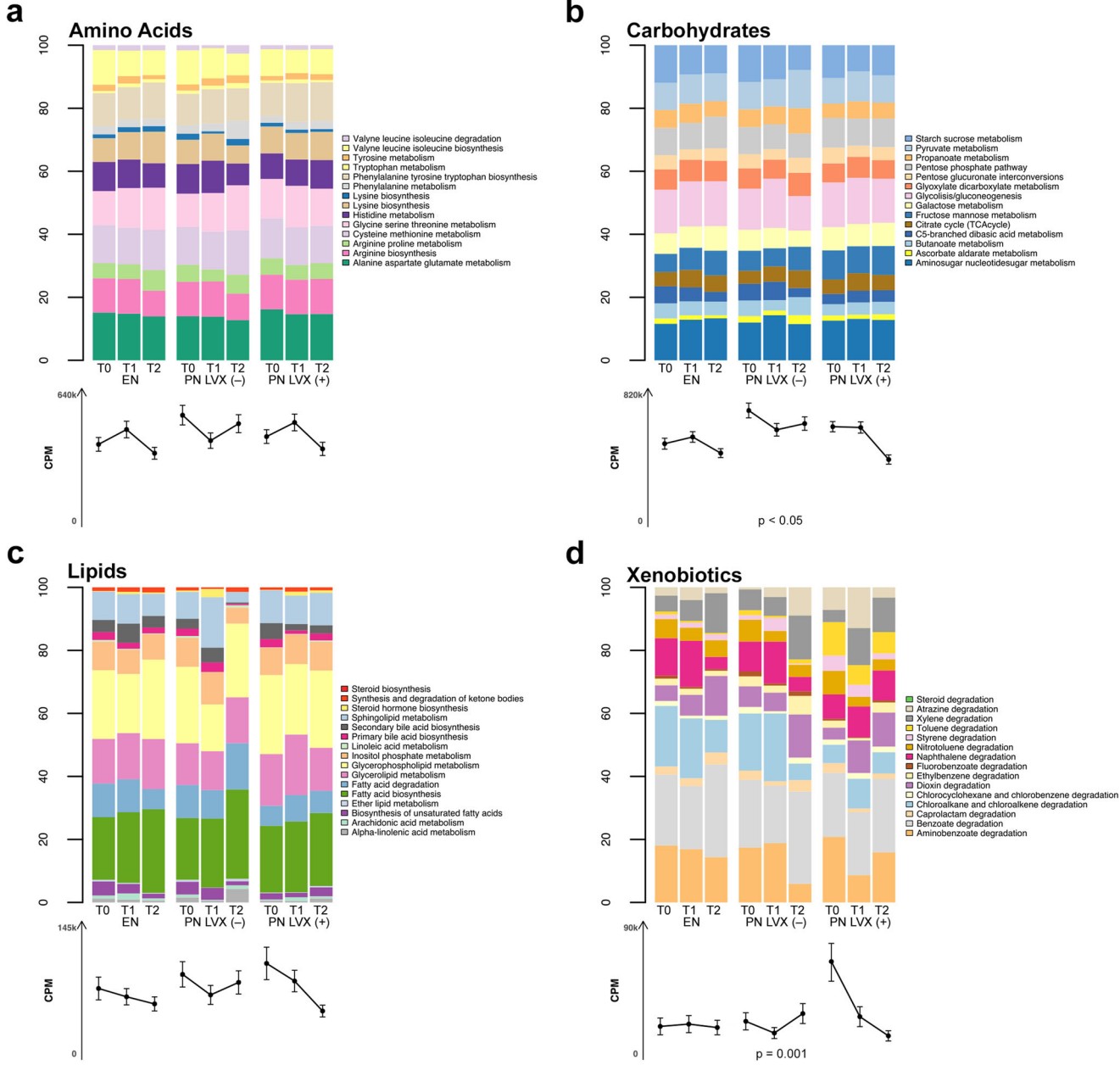

**Fig. 4 The gut microbiome profiles of pediatric HSCT patients receiving PN and LVX prophylaxis are associated with altered carbohydrate and xenobiotic metabolic potential.** On top of each figure, bar plots show the KEGG pathway classification of identified Kos divided by amino acid (**a**), carbohydrate (**b**), lipid (**c**), and xenobiotic (**d**) metabolism, represented as the mean relative contribution of each pathway to the total CPMs (copies per million) assigned to a given metabolism. Below each bar, the average number of normalized reads assigned to each metabolism is represented (CPM ± standard error of the mean). Data are shown for each patient group (PN combined with LVX prophylaxis—PN LVX (+) vs PN alone—PN LVX (−) vs EN alone—EN) before (T0) and at 2-time points after HSCT (T1 and T2). Significant differences among groups are shown (Kruskal–Wallis test with FDR correction). $n = 10$ biologically independent samples for each time point.

LVX (+) patients were overall distinguished by higher relative abundances of the *Klebsiella* triad and potential pathobionts *R. gnavus*, *F. plautii* and *E. faecium*, while lower proportions of typically health-associated taxa, including *F. prausnitzii* and bifidobacterial species.

**Local networks and the deconstruction of PN-related consortia.** To better disentangle gain-and-loss dynamics, we focused on the local network level, computing a correlation matrix and then constructing a network for each patient group at each timepoint. Three network parameters were adopted to describe the stability and interactions of the GM: modularity (i.e., the measure of

connections between and within modules), TC, and the ratio of negative to positive cohesion (N:P)[42,43]. In short, high modularity values indicate that the modules of a network are poorly associated with each other (i.e., more independent), therefore the impact of a stressor on a node (i.e., a taxon) is likely to affect only the members of its module, and not spread to the whole network. TC reflects the overall cohesive density of a network, with high values denoting higher intra-community crosstalk and presumably increased plasticity. Finally, low N:P values are indicative of a reduction in negative connectedness, which has been associated with high environmental stress[43], most likely reflecting the difficulty of continuing to maintain competing

**Table 1 List of identified hub species and keystone species in the gut microbiome networks of pediatric HSCT patients.**

| Treatment group | Node property | T0 | T1 | T2 |
|---|---|---|---|---|
| EN | Hub | Clostridium innocuum (1.60), Faecalibacterium prausnitzii (0.94), Eggerthella lenta (0.71) | Bifidobacterium longum (4.17), Bacteroides fragilis (0.3), Enterococcus faecalis (0.262), Bacteroides uniformis (0.2) | Bifidobacterium breve (6.60), Bifidobacterium longum (4.17), Ruthenibacterium lactatiformans (3.35), Parabacteroides distasonis (0.2) |
| | Keystone | Bifidobacterium breve (−58.89%), Bifidobacterium longum (−57.72%), Faecalibacterium prausnitzii (51.51%), Eggerthella lenta (−52.8%), Ruthenibacterium lactatiformans (−51.6%) | Eggerthella lenta (−53.37%), Bacteroides fragilis (−54.29%), Clostridium innocuum (−53.83%), Erysipelatoclostridium ramosum (−53.7%), Bacteroides vulgatus (−51.2%) | Bifidobacterium longum (−53.99%); then slightly lower than threshold: Erysipelatoclostridium ramosum (−49.75%) |
| PN LVX (−) | Hub | Akkermansia muciniphila (2.32), Bifidobacterium adolescentis (1.37), Faecalibacterium prausnitzii (1.29), Eggerthella lenta (0.19), Alistipes finegoldii (0.1) | Bifidobacterium longum (4.07), Bifidobacterium adolescentis (1.37), Eggerthella lenta (0.71) | Ruthenibacterium lactatiformans (3.35), Bacteroides vulgatus (1.23), Erysipelatoclostridium ramosum (0.5), Bacteroides uniformis (0.31) |
| | Keystone | Bifidobacterium longum (−57.04%), Bifidobacterium breve (−54.18%), Bifidobacterium adolescentis (−51.85%), Escherichia coli (−50.54%), Bacteroides uniformis (−50.39%) | Akkermansia muciniphila (−55.21%), Bifidobacterium breve (−51.57%) | Bacteroides fragilis (−51.03%), Bacteroides uniformis (−50.03%) |
| PN LVX (+) | Hub | Bifidobacterium adolescentis (1.37), Flavonifractor plautii (0.95), Fusicatenibacter saccharivorans (0.92), Roseburia inulivorans (0.2) | Clostridium innocuum (1.37), Klebsiella quasipneumoniae (1.02), Bacteroides dorei (0.86), Clostridium symbiosum (0.1) | Ruthenibacterium lactatiformans (3.55), [Ruminococcus] gnavus (2.15), Flavonifractor plautii (0.95), Enterococcus faecium (0.91) |
| | Keystone | No species detected whose absence caused a reduction >50% in TC. The more impactful was: [Ruminococcus] gnavus (−13.87%) | No species detected whose absence caused a reduction >50% in TC. The more impactful were: Akkermansia muciniphila (−25.5%), Klebsiella pneumoniae (−14.99%), Klebsiella quasipneumoniae (−13.49%), Klebsiella variicola (−13.85%) | No species detected whose absence caused a reduction >50% in TC. The more impactful were: Klebsiella pneumoniae (−24.99%), Klebsiella quasipneumoniae (−23.49%), Klebsiella variicola (−18.85%) |

Hub species and keystone species are reported for each patient group (PN combined with LVX prophylaxis—PN LVX (+) vs PN alone—PN LVX (−) vs EN alone—EN) before (T0) and at two time points after HSCT (T1 and T2). Hub nodes were detected using three measures of centrality, i.e., node degree, betweenness centrality, and closeness centrality. Only nodes with values of all three parameters greater than the median estimated value of their normal distribution were considered hubs. Relative abundance values are shown in the corresponding brackets. Keystone taxa were defined as such when their absence caused at least a halving of the total cohesion (TC) values in the network. A leave-one-out approach was adopted for one species at a time to check the impact of its absence over the microbial network. Percentages of reduction caused over TC values are shown in the corresponding brackets. Taxa scored as both hub node and keystone species for a given treatment group and timepoint are underlined.

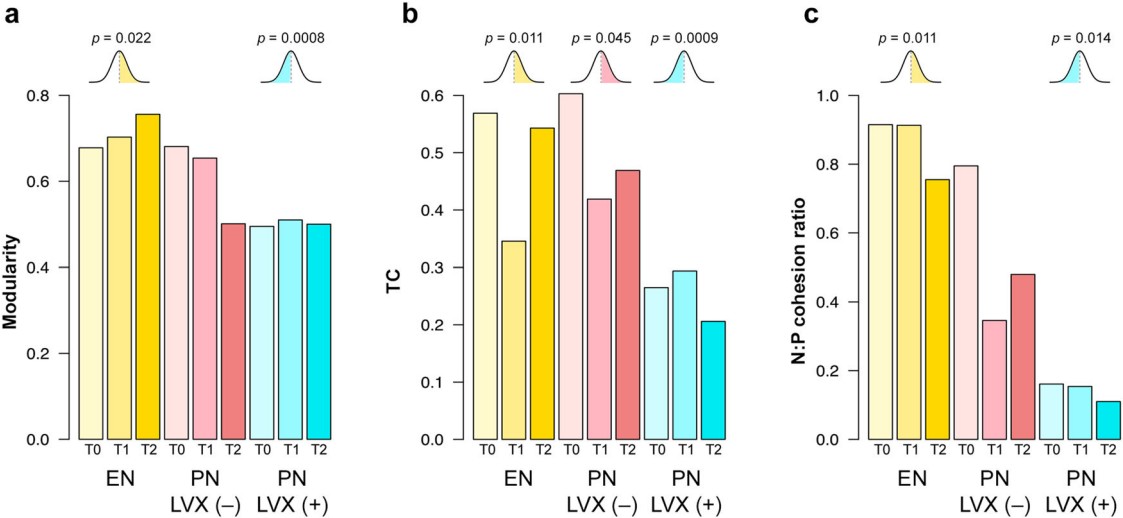

**Fig. 5 Network parameter analysis showed lower gut microbiome ability to withstand stress after LVX prophylaxis and PN in pediatric patients undergoing HSCT.** Each bar refers to a single network that was computed for the gut microbiome of each patient group (PN combined with LVX prophylaxis—PN LVX (+) vs PN alone—PN LVX (−) vs EN alone—EN) before (T0) and at 2 timepoints after HSCT (T1 and T2), leading to the derivation of a single parameter value for each network. The following network parameters were considered: modularity (i.e., the measure of connections between and within modules), total cohesion (TC, i.e., quantification of connectivity in terms of positive and negative interactions), and the ratio of negative to positive cohesion (N:P cohesion ratio). On top of each series of values a bell-curve represents the one-sample $t$ test performed and the corresponding one-sided $p$ value. See also Methods. $n = 10$ biologically independent samples for each timepoint.

interactions and resulting in an increased rely on cooperative interactions.

At baseline, the network structure of EN and PN LVX (−) patients showed greater modularity, higher TC values and a higher N:P ratio compared to PN LVX (+) ones (Fig. 5), thus suggesting a greater overall resistance of GM to stressful conditions in the absence of prophylaxis. Modularity remained unchanged over time for EN patients, while it decreased at T2 in the PN LVX (−) network, to values similar to those associated with LVX prophylaxis. Conversely, TC decreased at T1 and then recovered at T2 for EN and PN LVX (−) patients, while its values in PN LVX (+) patients remained stably low over time. Finally, the N:P ratio was markedly lower in the PN LVX (+) group than in the EN and PN LVX (−) groups at all timepoints, although a post-transplant decline was observed for the latter.

In an attempt to validate the observed trends, for each network parameter, we fitted all the computed values to a normal distribution and compared the scores of each group to the estimated median of the distribution using a one-sample one-sided $t$ test (see Methods). According to our findings (Fig. 5), the EN group showed significantly higher modularity than the estimated median of the distribution ($p = 0.022$), while the PN LVX (+) modularity was significantly lower ($p = 0.0008$). A similar trend was found for TC and the N:P ratio, with the EN samples falling above the median, while PN LVX (+) ones below ($p \leq 0.014$). As for TC, the PN LVX (−) samples also showed significantly higher values than the median ($p = 0.045$).

## Discussion
In this exploratory work, for the first time to the best of our knowledge we used an ecological networking approach to dissect GM network-wide signatures associated with LVX prophylaxis and nutritional support in pediatric patients undergoing HSCT. In particular, we coupled global and local networking scales, to exploit and complement the benefits of both approaches. The global network represents the entire networking structure

capturing all the relationship between nodes, including both direct and indirect associations. A local network is obtained considering the relationships between a clustered group of individuals according to—in our case—nutritional support, LVX prophylaxis and timepoint. The latter approach enables to more precisely focus on direct associations that are actually taking place between bacterial species in the considered cluster of patients, instead of focusing on all the possible associations that could take place in the cohort, thus facing the main weakness of global networking: the possible identification of notional modules based on indirect interactions rather than real syntrophic relationships. On the other hand, global networks are extremely useful to describe all the possible relationships between taxa in microbial communities and showcase to which extent each group of patients can populate the different modules or part of the network.

The global network structure we constructed, including all the samples, highlighted three modules, one of which containing what we called *Klebsiella* spp. triad (i.e., *K. pneumoniae*, *K. quasipneumoniae* and *K. variicola*, along with other pathobionts), and the other two comprising species generally present in the infant GM and associated with health (especially *Bifidobacterium* spp.). The latter were particularly underrepresented in PN patients, especially in those receiving prophylaxis, while the *Klebsiella* triad was overabundant in all PN subjects compared to EN. These data are not unexpected as the absence of enteral feeding is typically associated with the bloom of potential pathogens, such as *Klebsiella*, *Escherichia* and *Enterococcus*, to the detriment of beneficial commensals (including bifidobacteria and other SCFA producers), with the risk of an increased permeability, bacterial translocation, and therefore systemic infections[25,38,44]. In line with the available literature, the *Klebsiella* triad also included mucolytic species, namely *R. gnavus*, whose growth is known to be favored by the scarce availability of exogenous nutrients[45]. Supporting the global networking approach, the evaluation of significant differences in species-level relative abundances led to the identification of concordant signatures related to LVX administration.

When evaluating the main functions correlated to each species, we found that the *Klebsiella* spp. triad was strongly associated with pathways involved in the degradation of toluene and aromatic compounds, as well as the biosynthesis of L-ascorbate. Aromatic compounds and toluene potentially scavenged by the triad might be derived from the drug cocktails administered for the conditioning regimen[46]. Interestingly, L-ascorbate biosynthesis has been associated with the down-regulation of hematopoietic stem cell function and leukemogenesis, hence the role of this metabolic pathway would deserve to be specifically addressed also in the context of HSCT[47]. In contrast, some of the species most represented in EN patients (and typically health-associated), such as *F. prausnitzii*, *R. bromii*, *B. wexlerae*, *F. saccharivorans*, *G. formicilis*, were strongly correlated to pathways of starch degradation and methanogenesis starting from acetate, suggesting complete fermentation of carbohydrates with production of SCFAs. It should be remembered that SCFAs are microbial metabolites with multiple beneficial effects, which could play a strategic role in post-HSCT immunological reconstitution, reducing the risk of complications, such as infections and graft-versus-host disease, as widely discussed[21,48]. Furthermore, their fecal levels have already been shown to be rapidly restored in EN (but not PN) pediatric patients after transplantation, in parallel with a range of HSCT-related outcome benefits[25,36]. The aforementioned species were also correlated to the biosynthesis of several amino acids and MEP. The MEP pathway leads to the biosynthesis of isopentenyl diphosphate, a fundamental building block used by organisms in the biosynthesis of terpenes and terpenoids, a group of moieties that have been associated with intestinal anti-inflammatory responses, restoration of intestinal permeability, and regulation of inflammation signaling pathways in experimental models of inflammatory bowel disease through the prevention of oxidative stress[41]. Finally, we reported that xenobiotic degradative potential was significantly increased in PN LVX (+) patients, and shifted towards toluene degradation, particularly before HSCT, when LVX prophylaxis was already administered in almost all patients.

An overall better (i.e., healthier) GM network for HSCT patients fed EN was also confirmed by identifying hub nodes and keystone species. Taxa generally associated with health and well-known colonizers during childhood, such as *F. prausnitzii* and *Bifidobacterium* and *Bacteroides* spp.[49,50], were in fact scored as keystone or hub in EN at all timepoints. Some overlap was observed with PN patients who had not received LVX prophylaxis, although at T2 no bifidobacterial species were listed as hubs or keystones. In patients receiving prophylaxis and later PN, the GM network structure further deteriorated, with hubs including some of the pathobiont species mentioned above, namely *K. quasipneumoniae*, *E. faecium* and *R. gnavus*, as well as *F. plautii*. Interestingly, the latter has recently been associated with IgA deficiency in adults[51], and has been found to be enriched in children with irritable bowel syndrome[52] and autism spectrum disorders with abdominal pain[53], thus suggesting a possible involvement in dysbiotic layouts. It should also be noted that after HSCT, no hubs were shared between patients who received prophylaxis and those who did not, suggesting a potentially lasting negative impact of LVX administration on the network structure. The only exception is *R. lactatiformans*, which was identified as a hub node at T2 in all patients. Although no much information is currently available on this taxon, *R. lactatiformans* was part of an 11-strain consortium that induced interferon gamma (IFNγ[+])-producing CD8[+] T cells in mice, enhancing anti-cancer immunity[54]. Even with due caution, it is therefore conceivable that, as a central species in GM networks, it may play an important role in the modulation of the immune system (and in the efficacy of chemo-immunotherapy) also in the HSCT context. Furthermore, in patients receiving prophylaxis and

subsequently PN, we were unable to identify keystone species, since the absence of any species showed to reduce TC by no more than 25.5%, as found with *A. muciniphila*, followed by the *Klebsiella* spp. triad (range, 24.99%–13.49%) and *R. gnavus* (13.87%). This suggests extreme vulnerability of the microbial network in patients fed with PN, especially when preceded by LVX prophylaxis, with poor connectivity between microbial species and arguably a low ability to respond to external stressors.

The local network evaluation confirmed the strong PN-related perturbation of the GM network topology, especially in the PN LVX (+) group, as evidenced by the low values over time of modularity, TC and N:P ratio. Such network parameters are indicative of high levels of stress causing a reduction in connections among network members and reducing the ecosystem—normally relying mostly on negative interactions—to be based only on cooperative interactions[43,55].

Taken together, our exploratory findings demonstrate that PN in pediatric patients receiving HSCT—especially when coupled with LVX prophylaxis—exerts a destructuring effect over the GM network that persists beyond one month after HSCT. This weakening of the GM network results in low modularity, poor cohesion, the loss of beneficial microbes as hub/keystone nodes and the establishment of a 'non-network' of pathobionts, which share indirect associations rather than syntrophic relationships and carry possibly harmful functionalities. Conversely, EN promotes an overall stable and resilient GM network, with typical infant gut colonizers driving the community structure, which retain the potential to degrade dietary components and provide health-promoting metabolites (e.g., SCFAs), even in the stressful context of HSCT. Given the relevance of GM in HSCT outcomes and more generally in child health[21,56,57], the present results discourage the use of LVX prophylaxis and PN in order to preserve GM eubiosis. The main limitations of our pilot study are the small cohort size, which among others makes it difficult to rule out possible short- to long-term complications such as fatal sepsis, and the use of other antibiotics in addition to prophylaxis, which may represent a possible bias. Future prospective studies in larger cohorts, possibly with other omics approaches, such as metatranscriptomics and metabolomics, and finer long-term temporal resolution, are needed to validate our findings and correlate them with clinical outcomes.

## Methods

**Study cohort description.** This retrospective cohort study included pediatric patients receiving HSCT at the Pediatric Transplant Unit of the IRCCS Azienda Ospedaliero-Universitaria di Bologna, Italy, for whom DNA from at least three fecal samples was available. 30 eligible patients were selected from those previously reported in Biagi et al.[18] and D'Amico et al.[25]. In previous studies, feces were collected at different timepoints, before the conditioning regimen (i.e., T0), at the neutrophil engraftment (T1), as well as starting from 25 days after HSCT (T2). At our institution, patients admitted before May 2016 received antibiotic prophylaxis with I.V. LVX 10 mg/kg/day from the beginning of conditioning to the neutrophil engraftment; after that period, the center's policy changed not recommending anymore the use of LVX prophylaxis. For this patient group, nine out of ten samples were collected at least 24 hours after starting LVX prophylaxis. Patients received empiric and pathogen-directed antibiotics after the onset of fever and based on blood culture results, respectively, following the institutional antimicrobial stewardship program. Since February 2018, our institution has changed the internal protocol for nutrition during HSCT, preferring EN via nasogastric tube as the first line. Patients who could not tolerate the nasogastric tube received PN. From previously enrolled patients, those receiving and not receiving LVX and EN or PN were selected and matched for HSCT characteristics. Informed consent was collected from each patient's guardian or parent. The study protocol was approved by the Ethics Committee of Sant'Orsola-Malpighi University Hospital (ref. number 19/2013/U/Tess).

**Library preparation and shotgun sequencing.** DNA libraries were prepared using the QIAseq FX DNA library kit (Qiagen, Hilden, Germany) according to the manufacturer's instructions. Briefly, the microbial DNA was quantified with a Qubit fluorometer (Invitrogen, Waltham, MA, USA), and 100 ng of each sample was fragmented to a 450-bp size, end-repaired, and A-tailed using the FX enzyme

mix with the following thermal cycle: 4 °C for 1 min, 32 °C for 8 min, and 65 °C for 30 min. Illumina adapter barcodes were attached through a 15-min incubation at 20 °C in presence of the DNA ligase enzyme. Libraries were then purified twice with AMPure XP magnetic beads (Beckman Coulter, Brea, CA, USA), and a 10-cycle PCR amplification was performed. After a further purification step as described above, the final library was obtained by pooling all samples at 4 nM equimolar concentration. Thereafter, sequencing was performed on an Illumina NextSeq platform using a 2 × 150-bp paired-end protocol as per manufacturer's instructions (Illumina, San Diego, CA, USA).

**Network construction, identification of hub nodes, and keystone species**. MetaPhlAn 3.0.10[58] was adopted to obtain the composition of GM starting from shotgun metagenomic sequences. Species-level compositional data were filtered to keep species with relative abundance >0.3% in at least 5% of samples. Subsequently, a Spearman's correlation matrix was obtained, and only significant (FDR-corrected $p < 0.05$) comparisons were kept for the global co-occurrence network construction. To build the global network, we used the "igraph" R-package v1.2.6[59], removing edges with $-0.3 <$ weight $<0.3$ and proceeding with the identification of modules using the spin-glass model with default spins, gamma, temperatures, and cooling factor parameters, adding the "neg" implementation, to consider both positive and negative weights, and the "simple" update rule. For the local network construction, we computed a specific correlation matrix network for each study group (EN vs PN LVX (–) vs PN LVX (+)) at each timepoint (T0, T1, T2), to reconstruct the temporal dynamics in relation to LVX prophylaxis and nutritional support. The "igraph" package was used for the local network building, removal of edges with weight lower than 0.3, and topology evaluation, i.e., for the computation of modularity, nodes degree, betweenness centrality, and closeness centrality. Such parameters were fitted to a normal distribution with the "fitdist" function from the "fitdistrplus" R-package v 1.1.6[60] and only nodes with betweenness, closeness centrality, and degree above the estimated mean of the distribution were considered to be hub nodes. Connectedness metrics and cohesion values were computed as described by Herren et al.[42]. To identify keystone nodes, we adopted a brute-force "leave-one-out" approach, by setting one species abundance at a time to 0 and then proceeding to re-evaluate the network. Given that keystones are defined by the number of overall interactions that depend on them, we considered a node as keystone when its removal caused a reduction in TC >50% of the TC computed without setting that node abundance to 0.

**Functional analysis**. To gather functional insights from the shotgun metagenomic sequences, we implemented the HUMAnN 3.0.0[58] pipeline. Reads per kilobases assigned to gene families were normalized to copies per million (CPMs) accounting for sample sequencing depth, in addition to the default gene length normalization. Path abundances were considered for the global functional analysis. A Kendall correlation was adopted to link pathway CPMs with species-level microbial composition. Only pathways that showed significant differences in CPMs among the groups (see next paragraph) and only species with relative abundance >0.3% in at least 5% of the samples were considered for correlation. Gene families in UniRef90 were converted into KEGG Orthology (KO) for the targeted functional analysis of amino acid, carbohydrate, lipid, and xenobiotic metabolism.

**Statistics and reproducibility**. All the statistical analyses and network computations were performed in R 4.0[61]. Spearman's co-occurrence matrix $p$ values were corrected for multiple comparisons with the Benjamini-Hochberg method[62]. To compare modularity, TC and N:P among groups, for each parameter all the computed values were fitted to a normal distribution, and a one-sample one-sided $t$ test was performed for each group (EN vs PN LVX (–) vs PN LVX (+)). The values obtained from the constructed network of each group were tested for the null hypothesis by passing the estimated median of the distribution of the considered parameter as the mu argument and specifying the alternative hypothesis ("greater" or "less") depending on the trend observed for such parameters. Only significant one-sided $p$ values ($p < 0.05$) were considered. To detect pathways differently represented among groups, a Kruskal–Wallis test followed by FDR correction was used. Only pathways with FDR < 0.05 were then included in the analysis. Kruskal–Wallis test followed by FDR correction was first used to compare CPM values and relative abundances of the KOs for the targeted functional analysis of amino acid, carbohydrate, lipid, and xenobiotic metabolism. Only significant $p$ values (FDR-corrected $p < 0.05$) were considered. Subsequently, for the metabolisms that showed significant variations, pairwise comparisons with Wilcoxon rank sum test were performed. The heatmap function from the "stats" 4.0.3 R-package was used to represent the correlation heatmap. The software Cytoscape[63] was used for the graphical representation of the networks and the R-package "ggplot2"[64] for the remaining plots. Alpha diversity values were computed using the Shannon index by the 'vegan' R package[65] over the entire taxonomic assignment output from MetaPhlan. Kruskal–Wallis test and pairwise Wilcoxon rank sum tests with Benjamini-Hochberg correction were performed to detect significant differences ($p < 0.05$).

**Reporting summary**. Further information on research design is available in the Nature Portfolio Reporting Summary linked to this article.

## Data availability

High-quality paired-end sequences generated from this study can be found in the NCBI Sequence Read Archive under the Project Accession code PRJNA914532. Numerical source data (raw data) used for graphs and charts are available on Figshare at the following DOIs: Fig. 1 - https://doi.org/10.6084/m9.figshare.21749753; Fig. 2 - https://doi.org/10.6084/m9.figshare.21749768; Fig. 3 - https://doi.org/10.6084/m9.figshare.21749777; Fig. 4 - https://doi.org/10.6084/m9.figshare.21749780; Fig. 5 - https://doi.org/10.6084/m9.figshare.21749786.

## Code availability

This study did not generate any new tools.

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

## Acknowledgements

This research was funded by "Fondazione Cassa di Risparmio in Bologna", grant number "2020CARISBO_MASETTI_R".

## Author contributions

Conceptualization: S.R., M.C., S.T.; patients enrollment and sample collection: D.L., E.M., T.B., M.L.F., D.Z.; investigation: M.F., F.D., M.B.; writing—original draft: M.F., F.D., S.T.; writing—review & editing: M.F., F.D., D.L., E.M., D.Z., A.Pr., M.C., S.T., R.M.; Funding Acquisition: A.Pe., P.B., R.M. All authors read and approved the final manuscript.

## Competing interests

The authors declare no competing interests
