## [Peer Review File · Communications Biology]

Reviewers' comments:

Reviewer #1 (Remarks to the Author):

The authors nicely show the alterations in GM related to enteral nutrition and FQ prophylaxis.

The manuscript is well written and needs only minor adjustments.

1) In line 41 please mention also the recent improvements in survival outcomes after transplant in recent recents (Penack O et al. How much has allogeneic stem cell transplant-related mortality improved since the 1980s? A retrospective analysis from the EBMT. *Blood Adv.* 2020. doi:10.1182/bloodadvances.2020003418).

2) In line 50 it would be helpful to include some additional references to mention more extensively the correlation between GM and GvHD, as available in literature:

- Jenq R et al. *Biol Blood Marrow Transplant.* 2015. doi: 10.1016/j.bbmt.2015.04.016. Intestinal Blautia Is Associated with Reduced Death from Graft-versus-Host Disease.

- Greco R et al. *Blood.* 2021. doi: 10.1182/blood.2020007158. Microbiome markers are early predictors of acute GVHD in allogeneic hematopoietic stem cell transplant recipients.

3) Lines 57-62: authors should better highlight that the use fluoroquinolone prophylaxis (FQ-P) is controversial also in adult neutropenic patients. Recently, in this context, the European Conference on Infections in Leukemia group (ECIL) decided to re-assess the impact of FQ-P (Mikulska M, Averbuch D, Tissot F, et al. Fluoroquinolone prophylaxis in haematological cancer patients with neutropenia: ECIL critical appraisal of previous guidelines. *Journal of Infection* 2018; 76:20–37). Experts suggested weighting the benefit of FQ-P decreasing BSI rates against the disadvantages of its toxicity and ecological changes.

4) An additional comment outside the manuscript would be that it would significantly improve the value to look also at GM profile in patients (2/3 cases) receiving only FQ-P, without enteral/parenteral nutrition, comparing the results of GM in this group with the other findings. This would probably allow to better evaluate the contribution of FQ-P alone in this cohort.

Reviewer #2 (Remarks to the Author):

M.Fabbrini et al., described that LVFX prophylaxis for pediatric patients undergoing HSCT and claimed the combination of LVFX and parental nutrition would be disrupted gut microbiota stability.

It would be exciting and helpful, but the reviewer has a few questions.

It is very reasonable that antibiotics significantly affect intestinal bacteria.

However, among our separate antibiotic-free transplants, 7 of 10 bloodstream infections occurred in the parenteral nutrition, no LVFX group, and 0 bloodstream infections in the nasal feeding group.

What about the therapeutic use of non-quinolone antibiotics?

The diversity of T2 is likely to be significantly altered by interim antibiotics and should be described in detail.

For example, Kusakabe et al. (L592) described a good prognosis for a patient with a robust gut microbiome despite heavy antibiotic use.

It is somewhat difficult to understand that the transplant was done in the enteral feeding group with MAC transplantation and ATG use but no antibiotics at all. Therefore, it is helpful to describe detailed infection events and bloodstream infections.

Any other known differences between enteral feeding patients without bloodstream infection and those

with bloodstream infection should be described.

On the other hand, the possibility of fatal complications due to omitting gut decontamination should be described.

Since this cohort is a very small report, the possibility of fatal sepsis cannot be ruled out when this trial is carried out in a large-scale clinical trial.

Although carbohydrate and xenobiotic metabolism is described in Figure 3, it appears that the baseline differences are already quite significant in the T0 collection.

In the LVFX group, referring to Figure S1, is it possible that stool samples are collected after antibiotic use?

This suggests that the treatment is not homogeneous and should be evaluated more than the change between T0/T1/T2.

Reviewer #3 (Remarks to the Author):

In this manuscript the authors are investigate the impact of prophylactic antibiotic and nutrition support on the gut microbiota among pediatric HSCT recipients. This topic is clinically important and the authors' approach to describing the microbiome in this population using network analysis is novel.

Major Comments:

1) We have two exposures that the investigators are assessing, PN and levofloxacin, resulting in small numbers of patients in each group. The multiple comparison makes interpretation of results challenging

2) I would like to know more about how the authors are accounting for the time varying nature of exposures. At T0, all groups should not have been exposed to levofloxacin nor nutritional interventions. It is also possible that nutrition support occurred after T1 sampling. Are these observations categorized appropriately? A figure categorizing all antibiotic exposures and nutrition exposures during the follow-up time could be helpful.

3) I would also like to know more about the baseline comparison of specimens and the longitudinal within-subject comparison. It appears that the differences in microbiota exist at baseline, prior to receipt of levofloxacin or nutrition supplement. Thus it is difficult to determine the impact of levofloxacin or nutrition using a global approach.

4) Other antibiotic exposures are not described, but likely occurred. Levofloxacin prophylaxis has been shown to reduce exposure to other antibiotics by reducing neutropenic fever and bloodstream infections. In this population 70% of the LV- group had bloodstream infections compared to 30% in the LV+ group, suggesting that other antibiotic exposures occurred. Are we comparing levofloxacin prophylaxis to no antibiotic, or a combination of other empiric or pathogen-directed antibiotic regimens?

Minor Comments:

Abstract

Page 1 Line 22-23: The statement "Post-HSCT parenteral nutrition is oftentimes the first line approach" – should clarify the first approach for what? Severe mucositis or GVHD of the gut? As written it suggests that all HSCT patients receive PN.

Page 1 Lines 32-33: The authors extend their conclusions to GM-related HSCT complications, although that link hasn't been made. In reviewing the supplement, there was a much greater rate of bacteremia in the no-prophylaxis group despite the impact on the microbiome. I would be cautious about overstating the findings.

Introduction

Page 2, Line 62: Please clarify: PN is the first-line approach for what? All HSCT, or only with severe GI disease/mucositis/GVHD?

Results

Page 3 lines 87-90. What explains the heterogeneity in levofloxacin use at your institutions? Is it protocolized or provider-dependent? That information may help to reveal additional confounders. Similarly, would discuss potential confounding by indication where patients with more severe GI disease may receive PN as opposed to EN.

Discussion

Page 6 Lines 196 – 198. The authors state that the network structure differed at baseline, and that they relate to antibiotic prophylaxis. However, T0 precedes the administration of prophylactic levofloxacin suggesting that other confounders, or the limited sample size have resulted in differential microbiome characteristics at baseline. This is apparent in Figure 4 where the between group differences are large, but the within group differences are very small, despite my understanding that LVX (+) T0 should not yet have received levofloxacin.

Has this method of analysis been validated for assessment of antibiotic exposure? Hypothetically, all taxa susceptible to the antibiotic should decrease promptly. This simultaneous loss of species would result in a matching correlation among all susceptible taxa, and thus no keystone species can be identified (i.e., no individual species can have greater than 50% reduction in TC).

Methods:

Page 10, Lines 323-324. What features determine the receipt of PN vs EN? There is often confounding by indication where patients with severe GI disease receive PN.

How was the sample size determined?

Reviewer #1 (Remarks to the Author):

The authors nicely show the alterations in GM related to enteral nutrition and FQ prophylaxis. The manuscript is well written and needs only minor adjustments.

We thank the Reviewer for the time spent reading our manuscript and for the positive and constructive feedbacks.

1) In line 41 please mention also the recent improvements in survival outcomes after transplant in recent recents (Penack O et al. How much has allogeneic stem cell transplant–related mortality improved since the 1980s? A retrospective analysis from the EBMT. Blood Adv. 2020. doi:10.1182/bloodadvances.2020003418).

The Reviewer suggested an interesting and relevant article; thus, we added a sentence in the manuscript (Lines 42-43).

2) In line 50 it would be helpful to include some additional references to mention more extensively the correlation between GM and GvHD, as available in literature:

- Jenq R et al. Biol Blood Marrow Transplant. 2015. doi: 10.1016/j.bbmt.2015.04.016. Intestinal Blautia Is Associated with Reduced Death from Graft-versus-Host Disease.

- Greco R et al. Blood. 2021. doi: 10.1182/blood.2020007158. Microbiome markers are early predictors of acute GVHD in allogeneic hematopoietic stem cell transplant recipients.

We thank the Reviewer for pointing out these two interesting articles, which were indeed added to the manuscript (Lines 51, 52).

3) Lines 57-62: authors should better highlight that the use fluoroquinolone prophylaxis (FQ-P) is controversial also in adult neutropenic patients. Recently, in this context, the European Conference on Infections in Leukemia group (ECIL) decided to re-assess the impact of FQ-P (Mikulska M, Averbuch D, Tissot F, et al. Fluoroquinolone prophylaxis in haematological cancer patients with neutropenia: ECIL critical appraisal of previous guidelines. Journal of Infection 2018; 76:20–37). Experts suggested weighting the benefit of FQ-P decreasing BSI rates against the disadvantages of its toxicity and ecological changes.

We agree with the Reviewer in considering controversial the use of FQ-P. In the Introduction section, we further argued the criticalities of FQ-P use by providing evidence from adult neutropenic patients as suggested (Lines 61-66).

4) An additional comment outside the manuscript would be that it would significantly improve the value to look also at GM profile in patients (2/3 cases) receiving only FQ-P, without enteral/parenteral nutrition, comparing the results of GM in this group with the other findings. This would probably allow to better evaluate the contribution of FQ-P alone in this cohort.

The analysis suggested by the Reviewer is certainly of interest. Unfortunately, in our cohort, none of the patients completed the transplant without the need of a nutritional support. In fact, in pediatric HSCT settings, nearly all patients require nutritional support due to the extensive use of myeloablative conditioning and the related difficulties in taking foods orally.

Nonetheless, in an attempt to address the Reviewer's concerns, we added a new figure, Supplementary Figure 4, which shows family-level composition and species-level significant differences between patients stratified by FQ (levofloxacin) prophylaxis and timepoint, regardless of the nutritional support received after HSCT. This new analysis basically confirmed the networking results, with LVX (+) patients being overall distinguished by higher relative abundances of the Klebsiella triad and potential pathobionts R. gnavus, F. plautii and E. faecium, along with lower proportions of typically health-associated taxa (e.g., F. prausnitzii and bifidobacterial species) (Lines 203-208).

Reviewer #2 (Remarks to the Author):

M.Fabbrini et al., described that LVFX prophylaxis for pediatric patients undergoing HSCT and claimed the combination of LVFX and parental nutrition would be disrupted gut microbiota stability.

It would be exciting and helpful, but the reviewer has a few questions.

It is very reasonable that antibiotics significantly affect intestinal bacteria.

However, among our separate antibiotic-free transplants, 7 of 10 bloodstream infections occurred in the parenteral nutrition, no LVFX group, and 0 bloodstream infections in the nasal feeding group. What about the therapeutic use of non-quinolone antibiotics?

We thank the Reviewer for raising this important question. The use of other antibiotics is certainly a confounding factor in this field, which is also difficult to explore. In the new version of our manuscript, we added Supplementary Figure 2 and Supplementary Table 2, with the temporal representation of the antibiotic classes administered ad interim and information regarding bloodstream infections (with positive blood cultures), respectively. In addition – as suggested in the following point raised by the Reviewer – we evaluated the impact of interim antibiotics over alpha diversity (see the new Supplementary Figure 3) and found no significant differences in our cohort related to antibiotic intake (see lines 123-132 and 436-440 and also the next point).

The diversity of T2 is likely to be significantly altered by interim antibiotics and should be described in detail.

For example, Kusakabe et al. (L592) described a good prognosis for a patient with a robust gut microbiome despite heavy antibiotic use.

We thank the Reviewer for pointing out this interesting aspect of the study and apologize for not having explored it adequately in the previous version of the manuscript. As mentioned above, we prepared the following new supplementary items: i) Figure S2, which shows the antibiotic classes administered ad interim and the duration of their administration; ii) Table S2, which lists information on bloodstream infections with positive blood cultures; and iii) Figure S3, which shows the impact of interim antibiotic administration on alpha diversity. Regarding the latter, we stratified the patients by number of different interim non-quinolone antibiotics administered between the date of transplant and the sampled timepoints, and evaluated the Shannon index among the groups generated. Both the Kruskal-Wallis test and the pairwise Wilcoxon rank sum tests highlighted no significant differences, even when considering uncorrected p-values. These results therefore suggest that the main variables considered in the manuscript (i.e., nutritional support and levofloxacin prophylaxis) – in the framework of the stressful situation of the conditioning regimen – are likely to be more related to microbial shifts than the interim administration of antibiotics based on an empiric and pathogen-directed institutional stewardship program. Please, see lines 123-132 and also the new main Figure 2, where we show the alpha diversity values in the three patient groups (EN, PN LVX (-), PN LVX (+)) at the three timepoints (T0, T1, T2).

It is somewhat difficult to understand that the transplant was done in the enteral feeding group with MAC transplantation and ATG use but no antibiotics at all.

Therefore, it is helpful to describe detailed infection events and bloodstream infections.

We agree with the Reviewer and apologize for not having clarified this aspect earlier. In fact, most patients received other antibiotics for therapeutic use during the study window, which are now shown (divided by class) in the new Supplementary Figure 2, along with the duration of their administration. Moreover, as mentioned above, we prepared Supplementary Table 2, where we detailed the information regarding bloodstream infections. See also lines 99-100.

Any other known differences between enteral feeding patients without bloodstream infection and those with bloodstream infection should be described.

The point raised by the Reviewer is extremely interesting. However, fortunately, we cannot perform this analysis considering that no patient in our enteral feeding group experienced bloodstream infections.

On the other hand, the possibility of fatal complications due to omitting gut decontamination should be described.

We thank again the Reviewer for raising the point on fatal complications due to omitting gut decontamination. In this regard, it has recently been acknowledged that systemic antibacterial prophylaxis and selective gut decontamination in pediatric hematologic patients undergoing HSCT are no longer recommended during the conditioning period (Lehrnbecher et al., 2020, doi: 10.1093/CID/CIZ1082; Groll et al., 2021, doi: [https://doi.org/10.1016/S1470-2045\(20\)30723-3](https://doi.org/10.1016/S1470-2045(20)30723-3); Ifversen et al., 2021, doi: 10.3389/fped.2021.705179; Taur et al., 2014, doi: 10.1182/blood-2014-02-554725). On the other hand, immediate IV administration of pathogen-directed antibiotics is recommended, depending on local resistance patterns, patient's colonization status and antibiogram results (Ifversen et al., 2021). Of course, the possibility of fatal complications cannot be ruled out in any case. As specified in the next point, we briefly discussed this aspect in the new version of the manuscript (see lines 340-342).

Since this cohort is a very small report, the possibility of fatal sepsis cannot be ruled out when this trial is carried out in a large-scale clinical trial.

We thank the Reviewer for the remark. We indeed specified that we cannot exclude the possibility of fatal sepsis among the limitations of our study in the Discussion section (Lines 340-342).

Although carbohydrate and xenobiotic metabolism is described in Figure 3, it appears that the baseline differences are already quite significant in the T0 collection. In the LVFX group, referring to Figure S1, is it possible that stool samples are collected after antibiotic use? This suggests that the treatment is not homogeneous and should be evaluated more than the change between T0/T1/T2.

We greatly appreciated the Reviewer's suggestion as it allows us to better explain our study design. The collection of T0 samples for 9/10 patients in the LVX group took place at least 24 h after the first IV administration of levofloxacin. In the revised version of the manuscript, we specified this information in the Methods section (Lines 356-362) and revised Supplementary Figure S1, which now shows the LVX prophylaxis administration window for each PN LVX (+) patient. With specific regard to metabolic classes, we rearranged the related text in the Results section, clarifying that the basal differences between groups (namely those in xenobiotic metabolism) were likely related to the fact that PN LVX (+) patients had already been receiving LVX prophylaxis for at least 24 h (Lines 149-158).

Reviewer #3 (Remarks to the Author):

In this manuscript the authors are investigate the impact of prophylactic antibiotic and nutrition support on the gut microbiota among pediatric HSCT recipients. This topic is clinically important and the authors' approach to describing the microbiome in this population using network analysis is novel.

We really thank the Reviewer for appreciating the novelty of our approach.

Major Comments:

1) We have two exposures that the investigators are assessing, PN and levofloxacin, resulting in small numbers of patients in each group. The multiple comparison makes interpretation of results challenging. *We totally agree with the Reviewer's concerns, the cohort size is actually small and performing multiple tests inflates the type I error rate. As for the latter issue, in an attempt to consolidate the results, we corrected all the statistical tests performed for multiple comparisons with the Benjamini-Hochberg method (see lines 124, 387, 428, 429, 432, 482, 491, 510). Regarding the small sample size, in addition to specifying the pilot nature of our study throughout the text (Lines 22, 28, 243, 329, 340), we specifically reported this limitation in the Discussion section (Lines 340-342).*

2) I would like to know more about how the authors are accounting for the time varying nature of exposures. At T0, all groups should not have been exposed to levofloxacin nor nutritional interventions. It is also possible that nutrition support occurred after T1 sampling. Are these observations categorized

appropriately? A figure categorizing all antibiotic exposures and nutrition exposures during the follow-up time could be helpful.

We apologize to the Reviewer for not being clear on the study design in the first version of the manuscript. At T0, the levofloxacin group (9/10 patients) had already been receiving LVX prophylaxis for at least 24 hours. As also requested by Reviewer #2, we clarified this point in the Methods (Lines 356-359) and Results (Lines 150-153) sections, and revised Supplementary Figure 1, which now shows the LVX prophylaxis administration window for each PN LVX (+) patient. As for the nutritional support, this occurred after transplantation, depending on patient's tolerance to solid food, nasogastric tube and severity of adverse effects such as mucositis. Consistent with this, the GM profile of EN LVX (-) and PN LVX (-) groups at the baseline was mostly similar. In this regard, as suggested by Reviewer #1, we also performed a new analysis considering only the effect of LVX prophylaxis over microbial composition, regardless of the nutritional support occurring after HSCT (see the new Supplementary Figure 4 and lines 203-208). Finally, concerning all antibiotic exposures, we added a new Supplementary Figure 2 with the classes of interim antibiotics administered before, during and after transplant (and the duration of their administration) and a new Supplementary Table 2 with information regarding bloodstream infections (see also lines 99-100).

3) I would also like to know more about the baseline comparison of specimens and the longitudinal within-subject comparison. It appears that the differences in microbiota exist at baseline, prior to receipt of levofloxacin or nutrition supplement. Thus it is difficult to determine the impact of levofloxacin or nutrition using a global approach.

Again, we apologize to the Reviewer for not being clear on this issue. At baseline the differences in the gut microbial composition between groups (reported as part of the new Supplementary Figure 4) were related to the LVX prophylaxis administration that occurred in the PN LVX (+) group for at least 24 hours before T0 sampling (for 9/10 patients). This was better clarified in the revised version of our manuscript (please, see the answer to point above, in addition to lines 149-158, 203-208, 296, 357-360 and revised version of Supplementary Figure 1).

4) Other antibiotic exposures are not described, but likely occurred.

The Reviewer is totally right. As also requested by Reviewer #2 and anticipated above, we prepared a new Supplementary Figure 2, showing the patient's intake of antibiotic classes during the pre- and post-HSCT window, and a new Supplementary Table 2 with information regarding bloodstream infections (Lines 99-100). In addition, we evaluated the impact of the use of interim non-quinolone antibiotics (administered between the transplant date and the sampled timepoints) over alpha diversity using the Shannon index (new Supplementary Figure 3). Based on our findings, the administration of interim antibiotics had not a significant impact over the gut microbiota diversity after transplant, unlike nutritional support or previous LVX prophylaxis. Please, see lines 123-132 and also the new Figure 2, where we show the alpha diversity values in the three patient groups (EN, PN LVX (-), PN LVX (+)) at the three timepoints (T0, T1, T2) (where instead we observed significant differences over time).

Levofloxacin prophylaxis has been shown to reduce exposure to other antibiotics by reducing neutropenic fever and bloodstream infections. In this population 70% of the LV- group had bloodstream infections compared to 30% in the LV+ group, suggesting that other antibiotic exposures occurred. Are we comparing levofloxacin prophylaxis to no antibiotic, or a combination of other empiric or pathogen-directed antibiotic regimens?

We would like to thank the Reviewer for this important consideration. Almost every patient in both LVX (+) and LVX (-) groups received other empiric and pathogen-directed antibiotics, administered after the onset of fever or based on blood culture results, respectively, following the institutional antibiotic stewardship program. We detailed all the antibiotic classes received by each patient (and the duration of their administration) in the new Supplementary Figure 2 and provided information on bloodstream infections (with positive blood cultures) in the new Supplementary Table 2. See also lines 123-132 and 356-360.

Minor Comments:

Abstract

Page 1 Line 22-23: The statement “Post-HSCT parenteral nutrition is oftentimes the first line approach” – should clarify the first approach for what? Severe mucositis or GVHD of the gut? As written it suggests that all HSCT patients receive PN.

We apologize to the Reviewer for the lack of clarity in this sentence. As suggested, in the revised version of our manuscript, we specified that post-HSCT parenteral nutrition is oftentimes the first-line approach for nutritional support in the neutropenic phase (Line 22). Actually, most of the pediatric patients undergoing HSCT require nutritional support considering the difficulties in feeding by mouth due to severe nausea, pain and mucositis. Moreover, while most recommendations suggest the use of enteral nutrition, many pediatric HSCT centers still use parenteral nutrition.

Page 1 Lines 32-33: The authors extend their conclusions to GM-related HSCT complications, although that link hasn't been made. In reviewing the supplement, there was a much greater rate of bacteremia in the no-prophylaxis group despite the impact on the microbiome. I would be cautious about overstating the findings.

We agree with the Reviewer and apologize for the overstatement. We toned down our claims (Lines 30-32 and 340-342).

Introduction

Page 2, Line 62: Please clarify: PN is the first-line approach for what? All HSCT, or only with severe GI disease/mucositis/GVHD?

As in the Abstract, we specified that PN “has historically been provided as first-line approach for nutritional support in the pediatric post-HSCT neutropenic phase” (Lines 21, 69-70).

Results

Page 3 lines 87-90. What explains the heterogeneity in levofloxacin use at your institutions? Is it protocolized or provider-dependent? That information may help to reveal additional confounders. Similarly, would discuss potential confounding by indication where patients with more severe GI disease may receive PN as opposed to EN.

We thank the Reviewer for raising such issues. The use of levofloxacin as antibiotic prophylaxis was the standard of care in our institution until May 2016. Considering the emerging literature (Lehrnbecher et al., 2020, doi: 10.1093/CID/CIZ1082; Ifversen et al., 2021, doi: 10.3389/fped.2021.705179) questioning the role of levofloxacin in patient clinical outcomes, we decided thereafter to no longer administer levofloxacin prophylaxis to patients undergoing HSCT in accordance with our antimicrobial stewardship program. Regarding the choice of the type of nutritional support, all patients received EN as first-line nutritional support since February 2018. The only reason for preferring PN instead of EN was the difficulty of the patients to tolerate the nasogastric tube. We better detailed this in the Methods section (Lines 356-362).

Discussion

Page 6 Lines 196 – 198. The authors state that the network structure differed at baseline, and that they relate to antibiotic prophylaxis. However, T0 precedes the administration of prophylactic levofloxacin suggesting that other confounders, or the limited sample size have resulted in differential microbiome characteristics at baseline. This is apparent in Figure 4 where the between group differences are large, but the within group differences are very small, despite my understanding that LVX (+) T0 should not yet have received levofloxacin.

As specified in the points above, levofloxacin administration had already been started for 9/10 patients in the LVX (+) group at T0 (for at least 24 hours), thus baseline effects of LVX on GM were expected. We clarified this issue in the revised version of our manuscript (lines 152-154, 296, 357-360, 546-547, 569-571) and revised Supplementary Figure 1 accordingly. However, if the Reviewer believes that such a figure could

be informative to the reader, we will be more than willing to move it to the main text.

Has this method of analysis been validated for assessment of antibiotic exposure? Hypothetically, all taxa susceptible to the antibiotic should decrease promptly. This simultaneous loss of species would result in a matching correlation among all susceptible taxa, and thus no keystone species can be identified (i.e., no individual species can have greater than 50% reduction in TC).

The Reviewer's comment is totally correct. To the best of our knowledge, this method of analysis is extremely novel and has not been tailored for antibiotic exposure assessment. We tried to exploit the knowledge and techniques derived from networking approaches from other fields to address the biological and medical problem of studying the gut microbiome in pediatric HSCT settings. We decided arbitrarily to opt for a 50% reduction of TC given that half of the forces shaping the ecosystem should represent a massive variation in its ecological structure. In any case, in order to try to consolidate the approach, we decided to complement methods relying on completely different logics, i.e., the identification of hub nodes with the use of topology centrality measures and the keystone leave-one-out approach based on cohesion, obtaining oftentimes concordant results.

This method is certainly far from being validated and standardized, but we believe we did our best to address the ecological investigation of the microbial community through networking approaches.

Methods:

Page 10, Lines 323-324. What features determine the receipt of PN vs EN? There is often confounding by indication where patients with severe GI disease receive PN.

We partially addressed this issue in the previous comment. Indeed, since February 2018 all patients who underwent HSCT received EN nutrition as first-line nutritional support, while patients who did not tolerate the nasogastric tube received PN (Line 361-362).

How was the sample size determined?

No sample size was predetermined but we included all samples available in our cohort matching the criteria of stratification for nutritional support and antibiotic prophylaxis. In order to clarify that this study was only exploratory, we modified several sentences in the manuscript (see lines 22, 28, 243, 329, 340).

REVIEWERS' COMMENTS:

Reviewer #1 (Remarks to the Author):

The authors have adequately replied to my previous comments. No further comments from my side.

Reviewer #2 (Remarks to the Author):

This second version of the paper is a great improvement, the authors are to be commended.